# On the Importance of Both Dimensional and Discrete Models of Emotion

**DOI:** 10.3390/bs7040066

**Published:** 2017-09-29

**Authors:** Eddie Harmon-Jones, Cindy Harmon-Jones, Elizabeth Summerell

**Affiliations:** School of Psychology, The University of New South Wales, Sydney 2052, Australia; cindyharmonjones@gmail.com (C.H.-J.); summerell.elizabeth@gmail.com (E.S.)

**Keywords:** emotions, approach motivation, cognitive dissonance, anger, cognitive scope

## Abstract

We review research on the structure and functions of emotions that has benefitted from a serious consideration of both discrete and dimensional perspectives on emotion. To illustrate this point, we review research that demonstrates: (1) how affective valence within discrete emotions differs as a function of individuals and situations, and how these differences relate to various functions; (2) that anger (and other emotional states) should be considered as a discrete emotion but there are dimensions around and within anger; (3) that similarities exist between approach-related positive and negative discrete emotions and they have unique motivational functions; (4) that discrete emotions and broad dimensions of emotions both have unique functions; and (5) evidence that a “new” discrete emotion with discrete functions exists within a broader emotion family. We hope that this consideration of both discrete and dimensional perspectives on emotion will assist in understanding the functions of emotions.

## 1. Introduction

The field concerned with the psychological study of emotion has recently witnessed much debate regarding the structure of emotion. In particular, scientists have argued over whether emotions are best described along dimensions of valence and arousal [1,2,3] or as discrete entities [4,5,6,7,8,9]. In the psychological study of emotions, these two major meta-theoretical perspectives are often contrasted against one another. One of these perspectives, which we refer to as a dimensional one, posits that the basic building blocks or elements of emotions are dimensions such as arousal and valence (positive to negative, which is often regarded as synonymous with approach to withdrawal). This perspective builds on evidence that suggests that even simple organisms such as worms possess basic approach/avoidance responses, and then it posits that in more complex animals, discrete emotions such as anger and fear emerged from these basic emotive processes coupled with cognitive appraisals of the self and environment [10]. For example, Lang and Bradley’s [10] model posits the dimensions of arousal and valence, which, when combined, form an appetitive and aversive emotive orientation (i.e., arousal determines the intensity of the emotive orientation, and valence determines the emotive direction, with pleasant being appetitive and unpleasant being aversive, emotively speaking). Evolutionarily, they trace this approach–avoidance orientation to organisms as simple as worms and suggest that these dimensions underlie important physiological and psychological responses in humans. The other major perspective, which we refer to as a discrete one, posits that the basic elements of emotions are discrete entities such as anger, fear, joy, and so on [5,6,11,12]. For example, Panksepp [11] proposed a list of primary process emotions based on research using direct brain stimulation techniques in non-human animals, and Ekman [6] proposed a list of basic emotions based on research using cross-cultural studies on the recognition of facial expressions of emotions (and later using other methods). Each primary process or “basic” discrete emotion is posited to evoke a specific response tendency that will address a specific evolutionarily important need (e.g., protection from harm by fear, rejection of harmful substances by disgust). Evidence does suggest that most mammals possess a set of discrete emotions, but this perspective would not be able to incorporate the emotive responses of simpler organisms. Whether the discrete emotions evolved from the basic dimensions of approach and withdrawal or whether the discrete emotions exist independently of approach and withdrawal is debated, for good reason. We simply cannot know how these processes evolved from simple to more complex animals. We take the position that this issue most likely will never be resolved by research on the psychology of emotions, and that it is not necessary to resolve in order to increase our understanding of the functions of emotions. Thus, we believe both perspectives may have much to offer in attempting to understand the psychology of emotions. In this article, we will discuss why we believe both perspectives, broadly speaking, have something to offer, particularly with regard to the functions of emotions.

To illustrate why we believe that both dimensional and discrete perspectives have value, let us present an analogy from outside of psychology. Birds can be categorised in a discrete fashion, as different species. Birds could also be meaningfully described in terms of dimensions. Bluebirds are small, whereas ostriches are large. Geese fly long distances, whereas doves are resident birds that stay close to home. Describing a bird in terms of its dimensions (size or the distance it flies) is valid and provides one way of understanding differences among birds, but does not negate that other important information is provided by categorising a bird discretely, as a member of a specific species. Similarly, the fact that emotions may be described in terms of dimensions (valence, arousal, motivational direction) does not preclude the value that can be gained from taking a discrete emotions perspective (fear differs qualitatively from disgust, even though both are negatively valenced and avoidance motivated). In the end, both perspectives are simply theoretical speculations that may ultimately be inadequate. What is most important is what new information (scientific evidence) is generated by the theoretical perspectives.

In this article, we review evidence that illustrates how taking a dimensional and discrete emotions approach to the study of emotions contributes to better understanding the structure and functions of emotions. We begin by defining the constructs that we use in this article. Then, we review evidence from several lines of research that suggest that consideration of both discrete and dimensional perspectives to emotion aid in understanding the complex psychological construct of emotion. Specifically, we review research that demonstrates: (1) how affective valence within discrete emotions differs as a function of individuals and situations, and how these differences relate to various functions; (2) that anger (and other affective states such as dissonance) should be considered as a discrete emotion but there are dimensions around and within anger; (3) that similarities exist between approach-related positive and negative discrete emotions and they have unique motivational functions; (4) that both discrete emotions and broad dimensions of emotions have unique functions; and (5) evidence that a “new” discrete emotion with discrete functions exists within a broader emotion family.

## 2. Defining Our Constructs

Following the recommendations of Izard [13], we begin by defining the constructs we use in this article. Regarding the definition of emotion, we agree with Izard [13] (p. 367), who wrote, “Emotion consists of neural circuits (that are at least partially dedicated), response systems, and a feeling state/process that motivates and organises cognition and action. Emotion also provides information to the person experiencing it, and may include antecedent cognitive appraisals and ongoing cognition including an interpretation of its feeling state, expressions or social-communicative signals, and may motivate approach or avoidant behavior, exercise control/regulation of responses, and be social or relational in nature.” Izard [13] (p. 368) goes on to state that “emotions have multiple and quite significant functions in motivating and focusing individual endeavors, social interactions, and the development of adaptive and maladaptive behavior.” Thus, as Izard [13] states, emotion is a complex psychological process comprised of many components that may not be evoked or related to each other in every situation.

Some of the most important components of emotions are affective valence, motivational direction, and arousal. Affective valence is a broad component referring to how positive or negative individuals, including non-human animals (we posit), evaluate their feeling state [14]. Most individuals evaluate joy as positive, and they evaluate anger, disgust, fear, and sadness as negative. However, individuals also differ in how positive (and negative) they find these states; we return to this issue later. Motivational direction is another broad component and it refers to whether the organism is inclined to go toward (approach) or away (avoid) from something [15]. Individuals also differ in these motivational inclinations [16]. Finally, arousal, another broad component, refers to subjective experiences of being aroused as well as physiological responses (e.g., activation of the sympathetic nervous system).

Often, some of these components are considered similar. For example, some theories regard affective valence to be tightly coupled with motivational direction, such that positive affect is associated with approach motivation and negative affect is associated with avoidance motivation [3,17,18]. Much recent research has challenged these conceptions, and we review some of this challenging evidence below. Arousal is often regarded as independent of affective valence but it is often conceptualised as similar to motivational intensity (i.e., the intensity of approach or avoidance motivation; [18]). Again, research has challenged this conceptualisation [19]. For instance, an organism can be aroused by a recent bout of exercise but unmotivated to do anything, or one can be aroused by a humorous film but be unmotivated. Arousal may often be correlated with motivational intensity, although these constructs are non-equivalent and separable.

A number of discrete emotions likely exist, and many languages have words to describe these discrete emotions. Joy, anger, disgust, fear, and sadness are common discrete emotions that can be described by many words in the English language. A number of perspectives on the structure of emotions refer to “basic emotions” such as fear, anger, and joy. Following others [5,20], we use the term “basic emotion” to name the common, underlying construct that holds together an “emotion family”, and define a basic emotion as *the label for the central, most prototypical elements of a set of similar emotions*. For example, when a sorting method was used to categorise emotion words, “rage”, “frustration”, “bitterness”, and “irritation” clustered in a family with the prototype of “anger” [20]. We consider the “basic emotion” to reflect the communality that underlies a set of similar but not identical emotions. That is, the feeling state that persons label “irritable” may be lower in arousal than the state that people label “rage”, and “hostile” may more mood-like whereas “mad” is a short-term state. But all of these are subsumed under an “anger” category [20]. Thus, we consider “anger” to be the basic emotion, because it is most prototypical and best captures the intersection of these related emotions. However, the emotions within an emotion family may also be “discrete emotions”, in that they have unique characteristics and may be distinguishable from one another. The discrete states to which individuals would assign these labels almost certainly differ on dimensions such as arousal, valence, and motivational intensity, among other differences, while all possessing some features that are anger-like.

Some discrete emotions may exist and some languages may not have words to describe these discrete emotions. For example, one can receive genuine lavish praise from another person and feel a discrete experience that may contain elements of embarrassment and happiness, but there is no one word that accurately describes this discrete emotional feeling in the English language. We make this point because all too often emotion researchers limit themselves to the study of emotions that are described with the most commonly used emotion words in a language. Below, we discuss some recent evidence on a less common “discrete emotion” that lacks a formal label but is identified with the slang term “the heebie jeebies.” We show how researching this emotion reveals insight into the functions of discrete emotions.

Furthermore, labels of emotion-states are not equivalent with those states. This is due partly to subcultural and idiosyncratic understandings of words. For example, in certain English-speaking subcultures the words “jealousy” and “envy” are used as synonyms. Additionally, the same word may have more than one definition. For example, “hate” is a term that may describe either an attitude or an emotion depending on the context, and “love” is a term that may refer to a mildly pleasant positive affect (“I love this ice cream”) or an affect relating to intense sexual and romantic attachment (“I love my girlfriend”). Thus any verbal label or self-report measure of an emotion can only be a distant proxy for the presumed underlying construct [21,22]. We assume that discrete emotions are real states with functions that can be examined, even though language is inadequate to fully characterise them.

## 3. Considering Affective Valence

Even within discrete emotions, we find different individuals and situations are associated with different evaluations (positive to negative) of the feeling state. We refer to individual differences as traits, and situational responses as states, as is commonly done in the psychological literature. Thus, the degree of affective valence, one of the primary dimensions in dimension models, varies as a function of individuals and situations (or trait and states). Although our research on evaluations or attitudes toward discrete emotions has involved assessment of self-reported attitudes in adult humans, we posit that these attitudes do not need to be conscious or deliberate, and that many non-human animals have attitudes toward emotional feelings. That is why they will exert effort to experience positive affective brain states and exert effort to avoid negative affective brain states, as Panksepp [11] has reviewed. 

In general, discrete negative emotions like anger, disgust, fear, and sadness are evaluated as negative by most individuals [14]. However, individuals do differ in how negatively they evaluate these discrete negative emotions and their evaluations or attitudes vary as a function of the discrete emotion under consideration [14]. That is, some individuals have extremely negative attitudes toward anger, whereas other individuals do not (but these attitudes towards the discrete negative emotions are rarely on the positive end of a bipolar attitude scale) [14]. In addition, individuals who have extremely negative attitudes toward anger do not necessarily have extremely negative attitudes toward fear [14]. In other words, attitudes toward discrete emotions are relatively independent of each other; they load on separate factors and are not highly correlated with each other [14]. This evidence suggests that a discrete emotions perspective is useful to understanding individual differences in the dimension of affective valence.

Attitudes toward discrete emotions may also serve important functions. For instance, attitudes toward discrete emotions predict emotional situation selection, such that, for example, individuals with more positive attitudes toward fear (more accurately stated, as less negative attitudes toward fear) are more likely to express interest in viewing fear-evoking stimuli [14].

Moreover, attitudes toward discrete emotions correlate in different directions with their associated trait emotions depending on whether the trait emotions are approach- or withdrawal-motivated emotions [14]. Specifically, attitudes toward approach-motivated emotions, such as anger and joy, correlate directly with the reported experience of associated trait emotions (i.e., anger and joy). In contrast, attitudes toward withdrawal emotions (fear, disgust) correlate inversely with the experience of associated trait emotions (fear, disgust; [14]). Conceptually similar results occurred when attitudes toward emotions were used to predict emotional reactivity to emotional stimuli. That is, individuals who scored higher in their liking for approach-oriented emotions experienced more of the emotions in response to joy- and anger-evoking stimuli, whereas individuals who scored higher in their liking for withdrawal-oriented emotions experienced less of the emotions in response to fear- and disgust-evoking stimuli [14]. Taken together, these results suggest that the dimension of affective valence differs as a function of discrete emotions, and individual differences in these attitudes toward discrete emotions predict state and trait emotions depending on the discrete emotion’s classification as an approach- or withdrawal-motivated emotion. A consideration of both discrete and dimensional conceptual views on emotions is needed to explain this complex pattern of results.

## 4. Considering Anger

### 4.1. An Introduction to the Structure and Functions of Anger

Another line of research that has illustrated the importance of adopting both discrete and dimensional theoretical perspectives to better understand the functions of emotions is work on anger. That is, anger is a discrete (and likely basic) emotion but emotion-related dimensions exist around and within anger. However, before we illustrate how studying anger has benefitted by adopting both a discrete and a dimensional theoretical perspective, we need to review some past literature on anger. Anger is often considered a negatively valenced emotion that is high in arousal [2]. Prior to the 1990s, most theoretical perspectives posited that all negative emotions were associated with avoidance or withdrawal motivation [3,17,18]. However, since the 1990s, evidence has accumulated demonstrating that anger is primarily associated with approach motivation. This evidence has been revealed in research on non-human animals, young children, and adults using a variety of research methods (for review see, [15]).

Anger is commonly defined as a negative emotion that occurs when someone else is perceived as being responsible for a perceived misdeed or harm to ourselves and it motivates a desire to punish the responsible party [23]. However, much evidence has suggested that anger results from a variety of events (e.g., [24]), particularly when considering that the anger family involves several variations of feelings (e.g., frustration, annoyance, irritation). Some of these “angering” events are non-social and do not lead to attempts to punish another person. Carver [25], for example, provided evidence that angry feelings result when participants fail at a task (see also, [26]). Thus, we conceptualise anger as a state that often (but not always, see below) occurs when expected and desired approach-oriented goals are blocked [27]. This state of anger then functions to increase approach motivation that may cause movement towards the lost goal as well as other associated outcomes. This more abstract definition would encompass the anger we may feel when a vending machine fails to deliver the candy that we bought as well as the anger we feel when another person intentionally and unfairly cheats us out of money. Then, anger may motivate attempts to retrieve the candy by other means in the first instance, and it may motivate attempts to punish the thief in the second instance (e.g., punishment would be consistent with the approach-oriented goal of punishing the other to deter the person and others from theft in the future). Much empirical evidence suggests a link between approach tendencies and behavioural responses to anger (for review, see [28]). Indeed, human developmental research has revealed that anger is expressed on infants’ faces as early as two months when their positive reinforcements are extinguished, consistent with the idea that anger can result from non-social blocking of goal-directed activity [29,30]. 

However, approach-related anger occurs not only in response to disrupted approach, but a wide range of triggers [24]. Accordingly, the functions of angry approach are broader than simply achieving a positive outcome in which a lost object or goal is regained [28]. For instance, when surreptitiously instructed to adopt angry facial expressions, individuals show cognitive and physiological responses that suggest an increased approach-motivated state, even when the individuals are not aware they are creating angry facial expressions [31,32].

Angry facial expressions are often perceived as signalling behavioural approach intentions. For instance, direct eye gaze is associated with increases in the perceived intensity of facial expressions intended to communicate approach-related emotions, such as anger and joy [33]. Further, perceivers are faster to correctly detect approaching angry faces, compared to withdrawing angry faces, or approaching or withdrawing fear faces [34]. Interestingly, facial expressions intended to communicate anger are often confused with expressions of other approach motivated states, such as determination. More intense expressions of determination are often judged as appearing more anger-like [35].

Anger is also associated with approach motivation when measured at the reflex level. In one study, startle eyeblink inhibition, an approach motivational response, was associated with trait anger, enjoyment, and surprise during appetitive but not aversive images [36]. This finding suggests that trait anger is associated with approach motivation despite being experienced as a negative emotion.

Individual differences in approach motivation are associated with individual differences in anger-related traits. In adult humans, both state [25] and trait [37] anger have been found to correlate with approach motivation, as measured by Carver and White’s [38] trait behavioural activation sensitivity (BAS) scale. In children, mothers’ ratings of approach/positive anticipation and frustration/anger were positively correlated with each other, and with overt aggression [39].

Similarly, in humans, neural regions associated with approach motivation have been found to relate to anger. Higher resting levels of relative left (vs. right) anterior activity, a proxy for approach motivation, are associated with higher self-reported trait anger scores [40]. Moreover, activation in this neural region is greater in response to anger-inducing stimuli [41,42]. A meta-analysis found evidence that approach-related emotions, such as anger, are associated with greater left hemispheric activation ([43]; for a more recent review, see [44]).

In non-human animals, the relationship between anger and approach is particularly evident in the distinction between offensive and defensive aggression. Offensive aggression is typically associated with anger, and characterised by attack without attempts to flee [45]. In contrast, defensive aggression is typically associated with fear; it is associated with attempts to flee, and attack only occurs when the animal is unable to escape [46]. In an intruder paradigm, Lagerspetz [47] found that offensively motivated rats would endure pain, by crossing an electrified grid, to attack another rat. In contrast, defensively motivated rats would not, as their aggression functions to facilitate escape.

This distinction between offensive and defensive aggression in non-human animals extends to brain activation patterns. For example, rhesus monkeys have greater relative right frontal cortical activation and higher levels of cortisol during defensive aggression, but higher testosterone levels and lower cortisol levels during offensive aggression [48]. In rodents, the posteroventral medial amygdala and dorsomedial ventromedial hypothalamus are important in defensive aggression contexts, whereas the posterodorsal medial amygdala is important in offensive aggression contexts [49].

This relationship between anger and approach is also evident at the ‘trait’ level. Mice selected for high exploratory temperament displayed a range of more aggressive and approach-related behaviours compared to mice with a low exploratory temperament [50].

### 4.2. Exceptions to the Classifications of Anger as Negative and Approach Oriented

As reviewed above, anger is mostly negative and mostly approach oriented. However, anger is not inevitably negative and approach oriented, as recent research has revealed. This research suggests that even within a discrete emotion, the emotion can vary along the dimensions of valence and motivational direction. This idea that discrete emotions can vary along different dimensions was also proposed by Plutchik [51].

#### 4.2.1. Anger Associated with Withdrawal

Although primarily associated with approach, anger is also related positively to the behavioural inhibition system (BIS), at the simple correlation level [37]. The BIS is a motivational system that manages aversive motivation. Higher self-reported behavioural inhibition sensitivity, as measured by Carver and White’s BIS scale [38], is associated with greater relative right frontal cortical activity [52].

Research has found anger to be associated with relative right frontal cortical activity in certain situations. For instance, one study found higher self-reported feelings of anger were associated with relative right frontal cortical activity when individuals were instructed to mentally prepare for a situation in which the expression of anger was deemed socially inappropriate [53]. However, individuals who reported feeling angry also reported feeling anxious. Thus, anger may cease to be associated with approach when experienced in conjunction with other, withdrawal-related emotions. This research suggests the importance of considering mixed emotional states when examining the functions of emotions, as one emotion may inhibit the motivational characteristics of another.

Greater relative right frontal cortical activity has also been associated with angry rumination following an interpersonal insult. In one study, participants received transcranial direct current stimulation (tDCS) to increase either relative left or right frontal cortical activity, or sham stimulation [54]. Greater self-reported rumination occurred following a manipulated increase in relative right frontal cortical activity, but not following a manipulated increase in relative left frontal cortical activity or sham stimulation. Consistent with this result, angry rumination is associated with trait behavioural inhibition sensitivity [55].

Anger may also cease to be associated with approach when opportunities for approach-related action are blocked. For instance, participants showed greater relative left frontal cortical activity, a proxy for approach motivation, when they believed they could act to ameliorate an anger-inducing situation, but not when they believed they could do nothing [56]. However, participants reported equal anger in both conditions, suggesting that low coping potential blocked the motivational component of anger, but not the subjective experience. This suggests that consideration of the dimension of motivational direction is important in understanding the functions of a specific emotion within a situation.

However, Hewig and colleagues [57] failed to find that withdrawal-related anger related to relative right frontal activity. Anger-in, as measured by the State-Trait Anger Expression Inventory (STAXI; [58]), was hypothesised to relate to greater relative right frontal activation due to its conceptual similarities with BIS. However, results failed to support this prediction.

The research reviewed above suggests that anger may occasionally be associated with withdrawal rather than approach motivation. This research, however, has not clearly revealed the variables or conditions that cause anger to be associated more with withdrawal than approach motivation. The blocking of approach motivation (e.g., aggression) may be one reason anger is more associated with withdrawal than approach motivation, as suggested by Kelley et al. [54]. The threat of punishment may co-occur with conditions that also evoke anger and this threat may evoke fear and withdrawal motivation (e.g., Zinner et al., [53]).

#### 4.2.2. Anger Associated with Positive Valence?

Despite being typically conceptualised as a negative emotion, some evidence suggests that anger is associated with measures of positive valence. For instance, anger is directly related to *positive activation* (PA), a subscale of the Positive and Negative Affect Schedule (PANAS) [59,60]. In two studies manipulating anger, results indicated that anger was directly correlated with PA, and inversely correlated with happiness, despite PA and happiness correlating directly with each another [61,62]. Two subsequent studies replicated and extended these findings, and they also eliminated alternative explanations that either arousal or anger being experienced as a positive feeling could have caused the observed relationship [61]. In these studies, anger correlated even more strongly with PA when statistically controlling for happiness [61]. This relationship between anger and PA has been conceptually replicated in studies using trait measures [63].

One interpretation of these studies is that PA feelings may be evaluated positively by the person experiencing them while also being angry. It is also possible that, in an anger context, the PA items are not evaluated positively but are instead measures of approach motivation.

Motivational content is often obscured in self reports by the evaluative dimension of emotion (positive vs. negative valence), which appears more salient. For instance, when completing the “balanced PANAS”, a self-report scale comprising original PANAS items and antonyms of the same valence, responses tend to cluster strongly by valence, and not content [64]. However, after statistically controlling for the variance due to valence, anger items (*hostile, irritable*) cluster with approach-related positive items (*proud, excited, strong*). Thus, although evidence suggests a relationship between anger and approach, the prominence of the evaluative dimension means that this is not always immediately apparent (see also [61,63]).

## 5. Dissonance Affect and Approach Motivation

The research on anger has revealed both the importance of considering anger as a discrete emotion and the importance of considering dimensions around and within anger. The emotional response associated with cognitive dissonance also illustrates the importance of considering both discrete and dimensional views for understanding the functions of emotions. It also illustrates how there are dimensions around and within the affective state of dissonance. In brief, the emotive state of dissonance functions to enable the organism to resolve motivational conflicts so as to enact one of the motivations. We explain this functional view of dissonance in more detail after briefly explaining dissonance theory.

Cognitive dissonance is experienced when an individual holds two or more cognitions (elements of knowledge) that conflict with one another, such that one of the cognitions suggests that the other cognition ought not to be true. According to dissonance theory [65], this produces discomfort, which motivates the individual to do cognitive work to alter his or her cognitions to bring them more into alignment with each other. Dissonance occurs in a wide range of situations, as Festinger reviewed long ago. We give a few brief examples of situations that arouse dissonance. Dissonance occurs following difficult decisions because the positive aspects of the rejected decision alternative and the negative aspects of the chosen decision alternative are inconsistent with the decision that was made. To reduce dissonance, individuals often change their attitudes toward the decision alternatives, so that the chosen alternative is perceived as more positive and the rejected alternative is perceived as less positive than they were prior to the decision. Dissonance also occurs when individuals are exposed to information that is inconsistent with their beliefs or attitudes, and individuals in such situations can reduce dissonance in a variety of ways. Finally, dissonance occurs when individuals act contrary to their attitudes or beliefs (e.g., violate a moral standard) and the degree of dissonance that occurs depends on the reasons for the action (e.g., if individuals felt forced to act that way, they would feel less dissonance than if they did not).

Most perspectives on dissonance presume that dissonance-related cognitive changes are evoked by negative affect, and that attitude change reduces this negative affect [66,67], although affect is rarely measured in dissonance experiments. However, studies have shown that dissonance increases skin conductance, suggesting that it causes arousal [66,67]. Other research has shown that participants experiencing dissonance endorse items such as “uncomfortable”, “uneasy”, and “bothered” [68,69], suggesting that dissonance affect may involve vague, undifferentiated discomfort. Although Festinger [65] proposed the cognitive discrepancies should cause negative affect, he did not clearly state *why* this should be the case.

The action-based model is a reconceptualisation of dissonance that proposes a functional answer to the question of why holding conflicting cognitions may cause individuals negative affect [70,71]. This model proposes that cognitions serve as guides for behaviour, so when two cognitions are in opposition, their action tendencies are also likely to conflict. Thus, cognitive conflict may interfere with effective action. The negative affect of dissonance serves the function of motivating individuals to bring their cognitions into alignment so as to facilitate effective and unconflicted behaviour. The model further proposes that dissonance reduction should be an approach-motivated, action-oriented process that assists the individual in enacting recent decisions.

In support of this, studies found that individuals who were in an approach-motivated state as a result of an action-oriented mindset engaged in more dissonance reduction than those in a neutral or generally positive mindset [72,73]. Similarly, individuals who are high in trait approach motivation engage in more dissonance reduction [74], and greater dissonance reduction is associated with greater activation in the left frontal cortex [73,75,76], which is a proxy for approach motivation [44]. Furthermore, a manipulation that decreases approach motivation, the supine body posture, inhibits dissonance reduction [77]. Taken together, this research suggests that dissonance affect is an approach-motivated, negative affect that may serve the function of motivating cognitive changes that assist in reducing behavioural conflicts and enacting decisions.

We suspect that the subjective emotional feeling associated with dissonance may depend on the action tendencies (or cognitions) that caused the dissonance. That is, dissonance affect is generally a negative, approach-motivating emotional state, but it can be experienced as different discrete subjective states depending on the cognitions that caused the dissonance. A difficult decision may arouse vague feelings of negative affect, whereas morally violating a well-held standard may arouse feelings of guilt. In addition, although dissonance may typically be associated with approach motivation, particularly when it resulted from one’s own decision, it may be associated with withdrawal motivation in some circumstances. Indeed, when individuals are presented evidence indicating that they recently violated standards, they first experience guilt that is associated with withdrawal motivation [78]. However, when they are later presented with means of resolving the guilt, they evidence approach motivation [78]. These situations have been found to cause cognitive and behavioural responses attributed to dissonance, but, as mentioned previously, most past work on dissonance has not measured the subjective experience of dissonance.

## 6. Similarities between Approach-Related Positive and Negative Emotions: Anger and Determination

As reviewed above, much of the research that takes a dimensional approach to understanding emotion has focused on valence. The other dimensions of affect, such as motivational direction, have received less attention, although motivation may be highly relevant to the functions of emotion because it is more closely tied to behaviour. As we review below, similarities exist between some approach-related positive and negative discrete emotions and these emotions may have unique motivational functions. Thus, concurrently considering both discrete emotions and their similarities and differences on dimensions may lead to a better understanding of emotions with similar functions but different subjective evaluations.

A reliance on self-reports in studying emotion may have led to neglect of the action tendencies associated with emotions in favour of the subjective valence of emotion. As reviewed above, endorsements of emotion items rely strongly on an evaluative component that corresponds closely to valence [64]. A very different view of emotions may be found by examining perceptions of emotional facial expressions. From the perspective of the individual experiencing emotion, the most important dimension is its hedonic valence. The individual is highly aware of whether an emotion feels subjectively good or bad, and desires to maximise emotions that feel positive and eliminate emotions that feel negative. In contrast, perceivers of expressions of emotion are likely more concerned with the behavioural consequences of emotion, that is, predicting what the expresser is likely to do.

Examining emotions from the perspective of person perception instead of self-reports suggests that certain pleasant and unpleasant emotions may have similar behavioural tendencies. For example, determination is subjectively positive, whereas anger is subjectively negative. The word “determined” is one of the items on the positive affect subscale of the Positive and Negative Affect Schedule ([59]). This indicates that, in self-reports, determined loads on a positivity factor with other positive words such as “enthusiastic”, “interested”, and “inspired”. However, in early emotion research based on sorting facial expressions, determined and angry expressions were placed together at the *unpleasant* pole [79]. Furthermore, “determined” was one of the PANAS items that showed a reliable increase when state anger was evoked [61]. This suggests similarities between anger and determination, perhaps because both emotions are highly approach-motivated.

The idea that facial expressions of anger and determination appear similar was tested in several studies reported in one article [35]. In one experiment, participants were photographed while making voluntary facial expressions of anger, determination, joy, disgust, fear, sadness, and neutrality. These photos were shown to naïve judges, who attempted to identify the expressions using a forced choice procedure. Correct identifications of determination expressions were positively correlated with misidentifying the expression as anger, but not with misidentifying it as joy. In other words, the more identifiable a determination expression was, the more anger-like it appeared. In another study, highly identifiable determination, anger, and joy expressions were shown to participants, and they were asked to rate the intensity of determination, anger, and joy expressed in each photo. The intensity of determination was highly positively correlated with anger and negatively correlated with joy, such that photographs that were rated as more determined were also rated as more angry (*r* = 0.92) and less joyful (*r* = −0.85).

Anger and determination expressions likely appear similar because they have similar functions: Overcoming obstacles to reach a goal. Taking a purely discrete emotions perspective would not have revealed the overlap between these emotions, but considering the dimension of approach motivation inspired research that showed they are distinctly similar in their behavioural expression. In the future, applying a dimensional perspective to other discrete emotions may also uncover otherwise unexpected similarities and differences in their functions.

## 7. Functions Associated with Discrete Emotions vs. Dimensions of Emotions

Discrete emotions likely serve evolutionarily-derived psychological and behavioural functions [80,81]. For example, anger causes the organism to renew efforts to keep desired objects or it may cause destruction of the blocking agent [81] (p. 16). Sadness may cause the organism to reintegrate with socially supportive others, and fear may cause protection of oneself or others [81].

Dimensions of emotion, such as valence, arousal, and motivational direction, may have global psychological functions but are not likely involved in more discrete functions, such as the examples above. But some situations may be easily addressed by these more general functions. For example, research has revealed that emotionally arousing stimuli cause orienting, and that it is the broad dimension of arousal, rather than discrete emotions, that causes orienting [82].

In addition, emotions high in motivational intensity (desire, anger, fear) narrow cognitive scope, whereas emotions low in motivational intensity (satisfaction, some forms of sadness) broaden cognitive scope (for review see [83]). The narrowing of cognitive scope during motivationally intense emotions may serve the function of aiding in successful approach or avoidance. In other words, by focusing on the desired (or aversive) object, the organism may be more likely to obtain (or avoid) it; if the organism was not so focused and was distracted by other things, s/he might fail to obtain the desired object or avoid the harm. On the other hand, with emotions low in motivational intensity, organisms have reduced their efforts and the mind may broaden so that new opportunities can be seen. This process would be similar to the coasting in positive affect that Carver [84] has discussed and the disengagement from lost objects in sadness that several writers have discussed [85,86].

The first studies testing these broad ideas used positive affect varying in motivational intensity. In several studies using an attention task that measures the narrowness versus breadth of attention (Navon’s global-local letter task; [87]), high-approach positive affect led to more narrowed attention, compared to low-approach positive and neutral affect [88,89]. Moreover, higher trait levels of approach motivation predicted this narrowed attention [88].

This differential influence of motivational intensity has been found even when motivational intensity is manipulated using body postures. In one experiment, participants were instructed to smile while either leaning forward to induce a high approach positive state, sitting upright to induce a moderate approach positive state, or reclining backward to induce a low approach positive state [90]. The high approach positive state caused less inclusive categorisation on a cognitive categorisation task, compared to a low approach positive state. 

Other research has examined negative affect states varying in motivational intensity, yielding similar results. In one study, when negative affect low in motivational intensity was manipulated using sad images, participants showed a broadened attentional focus [91]. However, when negative affect high in motivational intensity was manipulated using disgusting images, participants showed a narrowed attentional focus [91]. Anger, a negative affect high in approach motivation, has also been found to narrow cognitive scope both perceptually and conceptually [92]. And decades of fear research showed similar patterns of narrowing [93].

An inherent question in this research on motivational intensity is the role of arousal; stimuli used to manipulate motivationally intense affect are also highly arousing. However, although motivational intensity and arousal are often associated with one another, they are often mistaken as equivalent. According to Bradley and Lang’s [94] theory of emotion, arousal is a proxy for motivational intensity, indicating “the degree of activation in each motivational system” (p. 585). But as mentioned previously, arousal is not always directly related to motivational intensity; they are separable.

To address arousal as a potential confound in research on cognitive scope, one study manipulated arousal independently of affect [19]. Participants pedalled on a stationary bike or sat still while viewing appetitive and neutral pictures and completing an attentional scope task. Attentional narrowing was greater following appetitive compared to neutral pictures. However, attentional scope did not differ between arousal conditions, despite those in the exercise condition having higher heart rates, indicating they were more aroused. Thus, arousal per se is not sufficient to cause the cognitive narrowing effect typically observed following manipulations of motivationally intense affect.

## 8. Discrete Emotions within Emotion Families

As the above sections demonstrate, considering dimensions, particularly approach versus avoidance, can assist with understanding the functions of discrete emotions. However, a discrete emotions perspective may also address specific functions of emotions more narrowly than is possible when considering dimensions.

For example, research on disgust and the “heebie jeebies” suggests that, even within an emotion family, different discrete emotions may have different functions. These functions, including subjective experience and behavioral tendencies, may not be fully captured by a focus on the broad dimensions of motivational direction and valence. From a dimensional perspective, disgust is avoidance-motivated and negatively valenced, as is fear, but it differs from fear in its subjective feel and behaviours. Disgust serves the function of motivating behaviours that help the organism to avoid pathogens [95]. However, recent research suggests that even within the disgust family there exists a unique, discrete emotion that is evoked by different stimuli and that motivates different behaviors than prototypical oral disgust [96]. This emotion does not have a standard label in English, but is identified by slang terms such as “the heebie jeebies”, “the willies”, or “the creeps”. Whereas oral disgust is evoked by stimuli that carry a high risk of disease if orally incorporated (feces, vomit, spoiled food), the heebie jeebies is evoked by stimuli that carry a risk of disease transmission via the skin (insects, skin lesions, reptiles). Typical disgust stimuli provoke sensations of nausea and an urge to spit something out of the mouth, whereas heebie jeebies stimuli provoke a skin-crawling sensation and an urge to protect the skin. Furthermore, the heebie jeebies showed some evidence of being a disgust-fear emotion blend. This may assist in understanding phobias, as animal phobias include elements of both fear and disgust, and may be better understood as reflecting the heebie jeebies [97]. Thus, although both the heebie jeebies and typical oral disgust have similar functions, as both motivate pathogen avoidance, they are associated with different stimuli, physical sensations, and behavioral responses. By simply taking a dimensional perspective, this would be missed, but even a discrete perspective that focuses on basic emotions does not capture all of the functions of unique emotions within emotion families.

## 9. Conclusions

We hope that this review has illustrated how a serious consideration of both dimensional and discrete conceptual views on emotions can enhance our understanding of the psychophysiology of emotions. Discrete emotions exist, as several reviews have proposed (e.g., [8,11]). Dimensions also exist. For example, dimensions of valence, arousal, and motivational direction may explain some global features of emotions and emotion-related processing, as we reviewed. And, as we also reviewed, dimensions likely exist within discrete emotions, which may assist in understanding varieties of discrete emotions. Ideally, these two broad theoretical models should generate solid research that aids in understanding the psychophysiology of emotions. We hope the research reviewed in this article contributes to a better understanding of emotions.

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
