# Peer review of "On the Importance of Both Dimensional and Discrete Models of Emotion"

_behavsci, 2017, doi:10.3390/bs7040066_

Round 1
Reviewer 1 Report
My basic response to this submission is that it illustrates the poverty of the dimensional approach, not its virtues. Reducing the behavior prompted by, using their example of anger, into approach and withdrawal is an unhelpful reductionism. Anger and fear are both approach, arousal, and negative emotions but are vastly different when considered functionally and behaviorally. A comparative, functional analysis dictates that anger leads to a tendency to threaten or attack, not simply to approach.
Adoption of the frustration-aggression hypothesis to explain anger is naive and dated. Reactions to frustration constitute anger only inconsistently; more common, as in animals, is simply trying another means of reaching the goal. More specifically, anger is elicited by intentional harm by someone, not by any failure to attain a goal, which can happen even in asocial situations where there is no one to attack. Similarly, a territorial animal will vigorously attack an intruder. This explains why accidental harm does not evoke anger even if it leads to frustration of our intention. Trivers' (1971) landmark analysis of "moralistic anger" shows how anger serves to enforce fair exchanges of favors. Current research on computer game playing has shown that individuals will sometimes incur a cost to themselves (frustrate themselves) in order to punish unfairness, to get angry. The function of anger, and the threat or attack that it can lead to, do not function to "destroy the blocking agent," but more precisely to punish the miscreant and deter future mistreatment. What is repeatedly lacking in this analysis is a real-world functional perspective, and one that implicitly recognizes that human emotions are similar to animal ones, and function in similar ways.
Observational research on humans and animals needs to be prioritized, not self-reports in lab settings involving artificial stimuli and no behavioral measure or even report of one's past behavior, only measures of judgments and attitudes. In particular, attitudes toward emotions are not essential facets of an emotion, and of little merit in understanding how emotions work. People's negative or positive attitudes toward anger miss the point of emotional valences. More productive would be to recognize that individuals differ in the intensity with which they various affects. This would offer a way of understanding personality differences as constituting differences in the threshold for, say, anxiety or anger. It is trivial to note that affects as a whole differ in intensity, valence (positive or negative), arousal, and approach-withdrawal. This is like recognizing that sensations can differ in intensity and complexity. You can't get very far with this; you have to get into the specifics of each sensation or emotion. For example, fear does not lead simply to withdrawal, or flight. It can also lead to hiding, seeking protection from others, freezing, death feigning, and defensive attack, depending on circumstances..Incidentally, they do not all involve arousal.
How do emotions work and what are their facets? An emotional affect is elicited by an internal or external elicitor and prompts a specific response tendency that addresses a specific fitness need, as for food, fluid, protection from attack, reproduction, etc. The behavior, in the case of some but not all emotions, is abetted by specific physiological adjustments. In some cases the affect also leads to an emotional expression that serves to alter the behavior of the receiver in ways that benefit the sender. Sometimes a given expression is evoked by more than one affect; for example, eye gaze conveys threat in some situations and friendly attention in others. It is not simply an approach signal. Joy and sadness likewise are not specific affects or expressions. Also, "determination" is a dubious basic emotion by, say, Ekman's criteria.
Another example of the simplicity of the dimensional approach is dividing anger into offensive and defensive types, citing Moyer. Actually, Moyer recognized several functional types of aggression: predatory, angry, defensive, inter-male (dominance), sexual, parental, and instrumental. These are distinct behaviors with different functions, elicitors, responses, expressions, and neural pathways.
Regarding neural mediation, the prefrontal cortex does other things besides mediating approach and attack. It also mediates higher-level problem solving, which may be going on in some of the research examples cited.
When the authors delve into the (indisputable) value of the discrete emotions approach, they illustrate its superiority. The analysis of cognitive dissonance is precisely what is needed--recognition of the affect underlying this phenomenon and its adaptive value. Likewise, the analysis of the observable behavior of disgust in functional terms is on point, especially recognition that disgust prompts expectoration of potential pathogens and not simply withdrawal. However, other sources of pathogens, such as snakes, corpses, and spiders, prompt not disgust but fear and avoidance. The functional approach offers a way to analyze pathological behaviors, namely as constituting aberrations from normal, adaptive behavior. The dimensional approach does not have this virtue.
Many of these points and others are made in the manuscript itself. I hope that these comments are constructive and lead the authors to reconsider their interest in the dimensional approach and instead turn their attention to analyzing specific emotions in functional terms.

Author Response
see the attached pdf for our reply

Reviewer 2 Report
This is an interesting article, and, with revisions, it is worth publishing. As it is, it needs some tightening up and there is at least one significant issue that, I think, the authors should consider changing.
I’ll start with the significant issue. The authors begin their article by saying, “scientists have argued over whether emotions are best described along dimensions of valence and arousal or as discrete entities. In our own research, we have studied emotions from a dimensional as well as discrete emotions perspective, as we believe that both perspectives have something to offer” (p. 1). There’s a terminological issue here (which, in other circumstances, might not be a big deal), but, in this case, I think that it’s interfering with what the authors are trying to say.
I, and I presume others, think of “dimensional” views of emotion as theories such as Plutchik’s multidimensional model where emotions are situated on a spectrum (or spectrums) of valence, arousal, or whatever. The authors of this paper don’t mean dimensional in that sense, however. Rather, they are proposing that emotions be viewed as either (1) simple entities, by which they mean that the entities have no parts, or (2) entities that are not simple. From this second perspective, each emotion type (anger, fear, sadness, etc.) can vary in its valence, arousal, or motivational direction. By looking at emotions in this way, the authors consider individual differences for certain emotions and how emotions can share some parts but not others.
Granted, the term “dimensional” can be used for the second perspective (the one where we acknowledge the emotions’ parts). Those parts do, after all, exist on dimensions. But the authors should be clearer about what they mean by dimensional if they are going to use that term. “Discrete” also isn’t really the best term to use since, whichever perspective they take, they are always referring to the emotions as discrete units (and never as, say, a spectrum where the terms fear, anger, sadness, etc. would not exactly apply).
* * *
Turning to some of the places where the article could be tightened up. Section 2 (“Defining our constructs”) should probably just focus on the concepts and ideas that the authors are going to use in the article. The quote from Izard in the first paragraph is probably too long to be just dropped into the article without more discussion. Plus, most of what’s in that quote isn’t really needed for the article. (And their own definition, given right before the quote (lines 50 – 51), isn’t really a complete definition.) I would also suggest making this section primarily about explaining the simple vs. non-simple perspectives that I mentioned above (and which the authors are going to use in subsequent sections).
Regarding the remaining sections, it would be helpful to preview (either in the first section or at the end of section 2 or beginning of section 3) what the plan is for the rest of the article and what will be covered. Section 4 is pretty interesting, but I found 4.1 confusing at first because I wasn’t sure what the point of it was.
Sections 5, 6, and 8 aren’t especially strong. They could be clearer and more detailed.
Section 7 is interesting, although the second paragraph (lines 361 – 363) is unclear to me.
Section 8 should, perhaps, be before 7. This section is interesting, although the authors don’t seem to state the main point, which is that “heebie jeebies” is also avoidance-motivated and negatively valenced. Also, a stronger case could be made here. I can see how disgust and heebie jeebies are similar except for some subtle difference, but, as it is, “avoidance-motivated and negatively valenced” is also true of, for instance, fear.
* * *
These are some more minor things that I noted while reading the article.
On lines 83 – 84, the authors write, “We use the term “basic emotion” interchangeably with “emotion family”, and define a basic emotion/emotion family as the label for the central, most prototypical elements of a set of similar emotions.” If what’s being labeled is just one emotion, then it’s not a family. On the other hand, if it is a group of similar emotions, then it sounds like there are some emotions that are more basic than the basic emotions.
The authors define affective valence as “how positive[ly] or negative[ly] individuals evaluate their feeling state” (p. 2, lines 61 – 62), and then they use that definition in the second paragraph in section 3. Although I agree that the emotions are generated by some sort of appraisal process, the valence has to be set before the subject makes a conscious, deliberative evaluation of the emotion, otherwise—assuming that the valence is part of the emotion—there wouldn’t be an emotion there to evaluate.
At the bottom of p. 3, the authors mention “state and trait emotions.” It would be useful to define those terms at some point.
On p. 3 (lines 117 – 118), the authors write, “In addition, individuals who have extremely negative attitudes toward anger do not necessarily have extremely negative attitudes toward fear,” and then they continue discussing that idea for a couple more sentences. There’s no citation there, however. Maybe the citation earlier in the paragraph is still supposed to apply? Related to that, throughout the article, there’s a notable lack of mentioning whose work or studies are being discussed (even when citations are provided). For a review article like this, it might help the organization and clarity of the article to include some names while discussing the different studies.

Author Response
see the attached pdf for our reply

Round 2
Reviewer 1 Report
First, I was unable to read the balloons because they extended beyond the margin of the page; I failed to figure out how to reveal them. Perhaps they rebutted some of my previous comments on the submission..
This may have been the case because very few of those comments seem to have resulted in changes in the manuscript. I believe that the authors pretty much rejected the evolutionary, functional approach that I favor. Because of that basic disagreement, I remain disappointed by the submission. I am attaching the last version of the paper with my previous comments so that I need not repeat them.
The authors actually begin by embracing that approach. They say that simpler organisms such as worms are capable merely of approach and avoidance behavior. I would say that this is itself a simplification, because even earthworms have to make different approach responses to, say, food, mates, and soil (feeding, mating, and burrowing). But it is reasonable that more complex organisms have more emotions in their behavioral repertoires.
Saying (line 150) that we cannot know how discrete emotions evolved is rather shocking to me. Comparative research on the brain (e.g., GE Schneider, Brain Structure and Its Origins), ethology, and genetics can and does reveal species differences and phylogenetic trends in emotional behavior. Eibl-Eibesfeldt argued that these taxonomic differences are an aid to understanding evolutionary trends. Yes, "we" cannot know about this unless we read the relevant literature, such as Detlier's classic To Know a Fly, which analyzes each and every neuron in this animal.
As I suggested last time, many of the statements in this manuscript are trivially obvious. Yes, emotions vary in intensity; so does every other psychological variable that is not all or none. Sometimes different terms are used to indicate differences in emotional intensity, such as "irritable" vs. "enraged," but these are just semantic distinctions and not indicative of qualitative differences. Little unites emotions that are intense; to understand them, we have to consider their discrete functions. The paper twists itself into knots by belaboring minor and laboratory-demonstrated similarities between, say, approach or arousing emotions. It's like the blind man probing the elephant.
The concept of arousal is vague, as the authors acknowledge on p. 3. It can refer to cortical activity, sympathetic division activity, or emotional excitement. Vagueness is not a good basis for developing a theory with explicit claims.
Likewise, approach is used in different ways in this paper, to refer to active engagement in behavior, attention, or approach to an object or another individual. And why complicate things by referring to a "Behavioral Inhibition System"?
The authors even use valence in different ways, to refer either to affect or to attitudes toward experiencing or displaying an affect, as mentioned previously.
On the other hand, in discrete emotions theory, the main concept is fitness enhancing behaviors, such as feeding, sex, sleep, defense, attack, and expectoration. These are explicit, valid across species, and functionally meaningful.
I am particularly disappointed that the authors cling to the discredited frustration-aggression hypothesis. I believe I cited Pastore's (1950) critique, which showed empirically that it wasn't any old frustration that elicited anger, but only a deliberate mistreatment. This also explains why we can be angered by someone else's being mistreated even though we are not ourselves frustrated. I also urged consideration of Hokanson's research on effects of mistreatment and how anger can be dissipated if the mistreatment is corrected in some way. Yes, we might kick a vending machine, but perhaps in order to punish the vendor who does not maintain his machines. These are effective, adaptive responses to address the specific misdeed. Also supporting this interpretation is Trivers's vital paper on reciprocal altruism and "moralistic anger" (1971). (Again, infants can't understand intentionality and so they are indeed angered by simple frustration.)
Sometimes obvious functional explanations are not noted. For example, a couple of cited studies showed that angry expressions are readily perceived. Well, detecting anger is a high priority for being safe. Lots of research shows that ambiguous expressions are often interpreted as angry. And the statement that "aggression functions to facilitate escape" (line 368) makes no sense. Again, lines 376f--mice that were highly exploratory as well as aggressive might simply have been low in fearfulness.
Another obvious assertion is that two emotions can come into conflict, such as fear and anger. It is not necessary to discuss approach in order to understand this.
As I wrote last time, I liked the functional explanation offered for cognitive dissonance; the authors even noted that Festinger failed to invoke one. But why not take one's own advice and consider the other findings in functional terms? For example, around line 575, the authors are perplexed by the fact that dissonance can be evoked by doing something that violates one's moral values. Well, that would lead to another behavioral conflict, between doing the right thing and catering to, say, one's popular prejudices.
Again I doubt that determination is a basic emotion. Does its expression prevail universally? What specific adaptive behavior does it induce? Does it have a specific neural basis? Does it occur in related species? The authors discuss basic, discrete emotions but do not lay out the criteria for a basic human emotion. Just because determination loads on a factor with other positive words means nothing; thousands of words would probably qualify.
Another obvious functional explanation (around line 642)--our attention narrows when we address a specific, urgent need such as for safety or mating. Our attention broadens when we confront no urgent fitness matter, and can devote our attention to acquiring general information for future use.
I do acknowledge the authors' adding reference to fear and phobias to the analysis of what they call the heebie jeebies. But I would disagree with the sentence on lines 704ff that a functional approach to discrete emotions is essentially limited; this example of refinement of functional analysis of disgust is a good example of how functional analysis that identifies different elicitors and behaviors for disgust and fear, respectively, validates this approach.
Minor comments:
line 87-don't need "and so on" after "such as"
line 201--again, "comprised of" is ungrammatical
line 268--animals don't have attitudes toward emotions, they can experience emotions in various degrees of intensity, such as weak or strong hunger...along these lines (ll. 283f), it is obvious that a person who has a positive attitude toward, say, fear is less afraid
line 329--this sentence is a non-sequitur; it interrupts the logical flow of the paragraph
line 357--add "cortical" after "anterior"

Reviewer 2 Report
In section 3, I gather that the idea is that self-reported attitudes are a measure of (or indication of) valence. If that’s right, I think that should be made clear in the first paragraph, and then the term ‘valence’ should be used instead of ‘attitudes’ in the remaining paragraphs in this section.
The second paragraph in section 4.1 includes some repetition that I would suggest fixing. (For example, in the first paragraph, there is “anger is often considered a negatively valenced emotion” and then at the beginning of the next paragraph there is “anger is commonly defined as a negative emotion.” Also in the second paragraph, there is “anger results from a variety of events” and in the third paragraph “anger occurs not only in response to disrupted approach, but [to] a wide range of triggers.” Of course, it’s okay to repeat things, but this has the look of repetition resulting from adding that second paragraph without modifying the rest of the section.)